# Silibinin Suppresses Tumor Cell-Intrinsic Resistance to Nintedanib and Enhances Its Clinical Activity in Lung Cancer

**DOI:** 10.3390/cancers13164168

**Published:** 2021-08-19

**Authors:** Joaquim Bosch-Barrera, Sara Verdura, José Carlos Ruffinelli, Enric Carcereny, Elia Sais, Elisabet Cuyàs, Ramon Palmero, Eugeni Lopez-Bonet, Alejandro Hernández-Martínez, Gloria Oliveras, Maria Buxó, Angel Izquierdo, Teresa Morán, Ernest Nadal, Javier A. Menendez

**Affiliations:** 1Medical Oncology, Catalan Institute of Oncology, Dr. Josep Trueta Hospital of Girona, 17007 Girona, Spain; esais@iconcologia.net (E.S.); ahmartinez@iconcologia.net (A.H.-M.); aizquierdo@iconcologia.net (A.I.); 2Department of Medical Sciences, Medical School, University of Girona, 17003 Girona, Spain; 3Girona Biomedical Research Institute (IDIBGI), 17190 (Salt) Girona, Spain; sverdura@idibgi.org (S.V.); ecuyas@idibgi.org (E.C.); mbuxo@idibgi.org (M.B.); 4Medical Oncology Department, Catalan Institute of Oncology, Hospital Duran i Reynals, 08908 L’Hospitalet de Llobregat, Spain; jruffinelli@iconcologia.net (J.C.R.); rpalmero@iconcologia.net (R.P.); enadal@iconcologia.net (E.N.); 5Medical Oncology Department, Catalan Institute of Oncology, Hospital Germans Trias i Pujol, 08916 Badalona, Spain; ecarcereny@iconcologia.net (E.C.); tmoran@iconcologia.net (T.M.); 6B-ARGO Group (Badalona Applied Research Group in Oncology), Germans Trias i Pujol Research Institute (IGTP), 08916 Badalona, Spain; 7Department of Anatomical Pathology, Dr. Josep Trueta Hospital of Girona, 17007 Girona, Spain; elopezbonet.girona.ics@gencat.cat (E.L.-B.); goliveras@iconcologia.net (G.O.); 8Hereditary Cancer Program, Epidemiology Unit and Girona Cancer Registry, Oncology Coordination Plan, Catalan Institute of Oncology-Girona Biomedical Research Institute (IDIBGI), 17007 Girona, Spain; 9Program against Cancer Therapeutic Resistance (ProCURE), Metabolism & Cancer Group, Catalan Institute of Oncology, 17190 (Salt) Girona, Spain

**Keywords:** nintedanib, non-small cell lung cancer, silibinin, STAT3, lysosome, lysosomal trapping

## Abstract

**Simple Summary:**

Nintedanib is an anti-angiogenic agent that has received approval in the European Union for the treatment of non-small cell lung cancer (NSCLC) after first-line chemotherapy. Here, we explore the possibility that the flavonolignan silibinin—the major bioactive component from the seeds of the milk thistle herb (*Silybum marianum*)—can provide clinical benefit in patients with advanced NSCLC treated with nintedanib. In vitro studies revealed that silibinin targets biological functions important for the therapeutic efficacy of nintedanib; specifically, activation of the transcription factor STAT3 and sequestration into lysosomal “safe houses”. Supplementation with the silibinin-based nutraceutical Legasil^®^ to patients with NSCLC receiving nintedanib/docetaxel was associated with increased clinical responses and a significantly longer time to treatment failure. Our findings provide a biological and clinical rationale for combining silibinin with nintedanib in a patient population for whom few effective second-line chemotherapy regimens are available.

**Abstract:**

The anti-angiogenic agent nintedanib has been shown to prolong overall and progression-free survival in patients with advanced non-small-cell lung cancer (NSCLC) who progress after first-line platinum-based chemotherapy and second-line immunotherapy. Here, we explored the molecular basis and the clinical benefit of incorporating the STAT3 inhibitor silibinin—a flavonolignan extracted from milk thistle—into nintedanib-based schedules in advanced NSCLC. First, we assessed the nature of the tumoricidal interaction between nintedanib and silibinin and the underlying relevance of STAT3 activation in a panel of human NSCLC cell lines. NSCLC cells with poorer cytotoxic responses to nintedanib exhibited a persistent, nintedanib-unresponsive activated STAT3 state, and deactivation by co-treatment with silibinin promoted synergistic cytotoxicity. Second, we tested whether silibinin could impact the lysosomal sequestration of nintedanib, a lung cancer cell-intrinsic mechanism of nintedanib resistance. Silibinin partially, but significantly, reduced the massive lysosomal entrapment of nintedanib occurring in nintedanib-refractory NSCLC cells, augmenting the ability of nintedanib to reach its intracellular targets. Third, we conducted a retrospective, observational multicenter study to determine the efficacy of incorporating an oral nutraceutical product containing silibinin in patients with NSCLC receiving a nintedanib/docetaxel combination in second- and further-line settings (*n* = 59). Overall response rate, defined as the combined rates of complete and partial responses, was significantly higher in the study cohort receiving silibinin supplementation (55%) than in the control cohort (22%, *p* = 0.011). Silibinin therapy was associated with a significantly longer time to treatment failure in multivariate analysis (hazard ratio 0.43, *p* = 0.013), despite the lack of overall survival benefit (hazard ratio 0.63, *p* = 0.190). Molecular mechanisms dictating the cancer cell-intrinsic responsiveness to nintedanib, such as STAT3 activation and lysosomal trapping, are amenable to pharmacological intervention with silibinin. A prospective, powered clinical trial is warranted to confirm the clinical relevance of these findings in patients with advanced NSCLC.

## 1. Introduction

Nintedanib (formally known as BIBF 1120) is an orally administered, broad-spectrum angiokinase inhibitor of growth factor receptors, including VEGFR1–3, PDGFRα/β and FGFR 1–4, as well as of FLT-3 and SRC non-receptor tyrosine kinases [1]. This multikinase inhibitor profile suggested that nintedanib could provide therapeutic benefit where other anti-angiogenic agents had failed. The pivotal phase III LUME-Lung 1 trial demonstrated that nintedanib plus docetaxel significantly improved progression-free survival compared with docetaxel alone in patients with advanced NSCLC treated with one line of platinum-based therapy [2]. Overall survival was also significantly extended in nintedanib/docetaxel-treated patients with adenocarcinoma histology, and a better disease control rate was observed in the patients in the nintedanib-containing arm despite a similar overall response rate in both arms. Based on these findings, nintedanib, in combination with docetaxel, was approved by the European Medicines Agency in November 2014 for the treatment of advanced NSCLC with adenocarcinoma tumor histology after first-line chemotherapy. Recent studies have supported the efficacy of nintedanib/docetaxel in patients with adenocarcinoma NSCLC after progression on prior chemotherapy followed by immune checkpoint inhibitor (ICI) therapy in real-world settings [3,4]. Despite the encouraging clinical data, however, the introduction of immunotherapy as second- and third-line treatment and the lack of new efficacy data on the nintedanib/docetaxel combination in daily clinical practice have largely discouraged lung oncologists from using this combination in off-trial settings.

In animal models of lung cancer in which, after initial nintedanib-induced regression, tumor growth resumes in the absence of active tumor revascularization, nintedanib treatment promotes either vascular normalization with hypoxia correction [5,6,7] or vascular pruning with elevated hypoxia [8,9,10,11,12,13,14]. Patients receiving neoadjuvant nintedanib show both normalization and increases in hypoxia, with the pattern of hypoxia correction correlating with tumor responses [15]. In this scenario, the question of whether the multikinase inhibition profile of nintedanib can directly affect tumor cell proliferation and survival becomes highly pertinent, as it might offer a tumor cell autonomous route to circumvent nintedanib via microenvironmental adaptive responses in a clinical setting. Intriguingly, the therapeutic activity of nintedanib extends beyond the targeting of angiogenic cell compartments [1] to involve not only fibroblasts in idiopathic lung fibrosis [16,17,18,19,20], but also cancer cells themselves. Indeed, single-agent nintedanib can directly inhibit cell proliferation and the survival of tumor cells in vitro—an effect that has been observed across a wide range of tumor types bearing a variety of cancer-driving genetic alterations [21,22,23,24].

The anti-proliferative/apoptotic activity of nintedanib on tumor cells seems to occur, at least in part, through the negative regulation of the signal transducer and activator of transcription 3 (STAT3) [24,25]. Nintedanib can operate as an agonist of the Src homology (SH2) domain-containing protein tyrosine phosphatase 1 (SHP-1), thus enhancing its capacity to dephosphorylate STAT3 at its activating tyrosine 705 (Tyr 705) residue [24,25]. Because over-activation of Janus associated kinase (JAK) or growth factor receptor-associated tyrosine kinase (SRC) contributes to the hyper-phosphorylation of STAT3, nintedanib might also inhibit STAT3 activity downstream of its actions on PDGFRβ, FGFR and SRC [26,27,28,29,30,31]. Importantly, ectopic overexpression and constitutive hyperactivation of STAT3 suffices to suppress the growth-inhibitory and apoptotic effects of nintedanib on cancer cells, revealing that the activation status of STAT3 [32,33,34,35,36] might be crucial in determining its non-angiogenic cancer cell-targeted activity.

Common strategies for inhibiting STAT3 signaling include indirect inhibition of upstream receptors (JAKs, IL-6R) or direct blockade of STAT3 dimerization by obstructing the STAT3 domain, impeding STAT3-DNA binding and inhibiting STAT3 transcription. Unfortunately, the majority of direct STAT3 inhibitors (STAT3i) have yet to enter clinical evaluation and, despite decades of research, very few FDA-approved STAT3i have emerged [37,38,39,40,41,42]. Using computational and experimental approaches, we recently delineated the ability of the flavonolignan silibinin—the major bioactive component in the silymarin extract obtained from the seeds of the milk thistle herb (*Silybum marianum*) [43,44,45,46] to inhibit STAT3 activity. Silibinin operates as a direct STAT3i by blocking the STAT3 SH2 domain, which is crucial for both STAT3 phosphoactivation and nuclear translocation, while also targeting the STAT3 DNA-binding domain, which is crucial to drive STAT3 transcriptional activity irrespective of its activation/dimerization status [47]. Such multi-faceted activity of silibinin against STAT3, when used orally as part of the bioavailable Eurosil-85^®^/Euromed nutraceutical formulation contained in the nutraceutical product Legasil^®^ [48], has proven therapeutically effective in patients with advanced NSCLC and was particularly notable in the central nervous system, where it provided greater than 4-fold survival benefit in patients with established brain metastases [49,50]. This groundbreaking clinical activity of silibinin was accompanied by low toxicity and reversible secondary effects and was compatible with the standard-of-care in oncology patients [51,52].

Here, we explore the molecular basis and the clinical benefit of incorporating silibinin into the nintedanib/docetaxel schedule in NSCLC. First, we assess the nature of the tumoricidal interaction between nintedanib and silibinin and the underlying impact on STAT3 activity in a panel of human NSCLC cell lines [53,54,55,56,57,58]. Second, we explore the impact of silibinin on additional molecular mechanisms that might confer cancer cell-intrinsic resistance to nintedanib such as sequestration inside the lysosomal “drug safe-house” [59,60]. Third, we conduct a retrospective, observational multicenter study to assess the efficacy of Legasil^®^ in patients with NSCLC receiving the nintedanib/docetaxel combination in second- and further-line settings.

## 2. Results

### 2.1. Inverse Correlation between Nintedanib and Silibinin Sensitivity in Human NSCLC Cell Lines

We first characterized the cytotoxic effects of nintedanib and silibinin as single agents against a panel of molecularly diverse NSCLC cell lines (*n* = 7) (Table 1). Cells were seeded in microtiter plates and then cultured without or with increasing concentrations of nintedanib or silibinin for 72 h. Metabolic status, as a measure of viability, was assessed by the conversion of the tetrazolium salt, MTT, to its formazan product. We observed a dose-dependent reduction in cell viability in response to both nintedanib and silibinin, but with evident differences in sensitivity between cell lines. To examine this more closely, we generated concentration-effect curves by plotting the fraction of unaffected (surviving) cells against nintedanib/silibinin concentrations and calculated IC_50_ values: for nintedanib, IC_50_ values ranged from 2.83 ± 0.45 µmol/L in PC-9 cells harboring an EGFR activating mutation (Δ746–750) to 9.31 ± 0.40 µmol/L in FGFR1-overexpressing H1975 cells harboring the EGFR T790M resistance mutation (Figure 1A); IC_50_ values for silibinin ranged from 52.58 ± 1.65 µmol/L in KRAS (Q61H)-mutant H460 cells to 122.13 ± 6.66 µmol/L in H2228 cells bearing the EML4-ALK E6a/b:A20 variant 3 (Figure 1A). We also observed a tendency for a negative correlation between the apparent degree of sensitivity of NSCLC cell lines to the drugs, with those more sensitive to nintedanib being less sensitive to silibinin, and vice versa. In terms of IC_50_ values, on average, cell lines more responsive to nintedanib were ~2-fold more resistant to silibinin, whereas cell lines more responsive to silibinin were ~3-fold more resistant to nintedanib.

### 2.2. Synergistic Interactions between Nintedanib and Silibinin in Human NSCLC Cells

To investigate whether silibinin co-exposure might improve the cytotoxic activity of nintedanib, we treated the same NSCLC cell lines with nintedanib in the absence or presence of fixed concentrations of silibinin (50 and 100 μmol/L). Likewise, NSCLC cell lines were exposed to silibinin in the absence or presence of fixed concentrations of nintedanib (2.5 and 5 μmol/L) to evaluate the sensitizing effects of nintedanib towards silibinin. To measure changes in sensitivity, sensitization factors were determined by dividing the IC_50_ values of nintedanib or silibinin as single agents by those in the presence of silibinin or nintedanib, respectively (Figure 1B). Results showed that silibinin enhanced the cytotoxic activity of nintedanib in a dose-dependent manner, which was particularly notable in the cell lines exhibiting intrinsically higher IC_50_ values for nintedanib (H460 and H1975, hereafter “nintedanib-resistant”). Indeed, treatment with 100 μmol/L silibinin increased the cytotoxic activity of nintedanib by >50-fold. In the converse experiment, nintedanib increased the cytotoxic activity of silibinin in a dose-dependent manner and this was particularly evident in those cell lines exhibiting intrinsically higher IC_50_ values for silibinin (PC-9 and A549, hereafter “silibinin-resistant”), with 5 μmol/L nintedanib increasing the cytotoxic activity of silibinin ~10-fold (Figure 1B).

As single agents, silibinin and nintedanib suppressed NSCLC cell viability at the concentrations employed as sensitization agents. Because the observed sensitization effect might be due to their own toxicity, potential synergistic interactions could not be accurately discriminated from additive or antagonistic effects based on these data alone. Although there is controversy over which method is best for detecting true in vitro synergy of drug combinations, we elected to analyze the impact of the nintedanib plus silibinin combination at different concentration levels, so that a constant molar ratio of nintedanib to silibinin was maintained. When the dose ratio of the drugs was fixed at 1:20 (nintedanib:silibinin), the extent of the cytotoxic interaction was again cell line dependent. The true nature of the interaction was then assessed using the Compusyn program (https://www.combosyn.com/; accessed on 1 June 2020) to calculate the CI parameter, which indicates whether the doses of the 2 agents required to produce a given degree of cell toxicity are greater than the doses that would be required if the effects of nintedanib and silibinin were strictly additive (Figure 1C, *left*). Combinations of nintedanib and silibinin generated CI values of 0.79 to 1.00 for IC_50_ and 0.49 to 1.04 for IC_90_, indicating additive effects in most cases. Of note, however, was the finding of synergistic interactions in those cell lines exhibiting intrinsically higher IC_50_ values for nintedanib (H1975 and H460) or to silibinin (PC-9 and A549). Such synergistic interactions were more evident at high levels of cell killing (IC_90_) (Figure 1C, *left*). Moreover, the nature of the interaction between nintedanib and silibinin remained largely unaltered in NSCLC cells chronically adapted to nintedanib by incremental and continuous exposure to the drug for >6 months (Figure 1C, *right*).

### 2.3. Synergistic Interactions between Silibinin and Nintedanib in NSCLC Cells with Acquired Resistance to EGFR- and ALK-Tyrosine Kinase Inhibitors

We questioned whether the generalized progression of EGFR-mutant and ALK-translocated NSCLC after an initial response to EGFR and ALK tyrosine kinase inhibitors (TKIs) in a clinical setting would alter the nature of the interaction between nintedanib and silibinin. To do this we used three cell lines: PC-9/ER cells, a pre-clinical model of acquired resistance to the EGFR TKI erlotinib established by growing PC-9 cells in the continuous presence of high doses of erlotinib, but lacking additional inactivating mutations in the EGFR kinase domain [55,56]; H2228/CR cells, a crizotinib-resistant variant obtained through long-term exposure to increasing concentrations of crizotinib that activates the epithelial-to-mesenchymal transition (EMT) program [54]; and H3122/CR cells, a crizotinib-resistant variant capable of growing in the presence of 1 µmol/L crizotinib due to activation of “bypass” alternative kinases (EGFR), but lacking amplification or mutations in the kinase domain of ALK [53]. In terms of IC_50_ values, responsiveness to nintedanib was slightly decreased in PC-9/ER cells, whereas a significant collateral resistance to nintedanib was observed in cell lines with acquired resistance to crizotinib (Appendix AA). In H3122/CR cells the IC_50_ value of nintedanib increased more than 2-fold. Conversely, we failed to detect a difference in sensitivity to silibinin in any of the EGFR-TKI and ALK-TKI resistant derivatives (Appendix AA).

Co-treatment with silibinin enhanced the cytotoxic activity of nintedanib in a dose-dependent manner, which was particularly evident in EGFR-TKI and ALK-TKI resistant cells (Appendix AB,C). Analysis of the cell viability data of the nintedanib plus silibinin combination from automated determination of CIs demonstrated that the synergistic interaction in erlotinib-responsive PC-9 cells remained unaltered in PC-9/ER cells (Appendix AD). Moreover, the nature of the interaction between nintedanib and silibinin switched from additive in crizotinib-responsive H2228 and H3122 parental cells to synergistic in the isogenic H2228/CR and H3122/CR derivatives with acquired resistance to crizotinib (Appendix AD).

### 2.4. Synergistic Interactions between Nintedanib and Docetaxel in NSCLC Cells

When we characterized the baseline cytotoxic effects of docetaxel as a single agent in the same panel of molecularly diverse NSCLC cell lines, we noted that sensitivity was lower in cell lines with primary resistance to erlotinib. H1993 cells bearing a MET amplification, which is recognized as one of the molecular mechanisms of EGFR-mutant NSCLC resistance to EGFR-TKIs, were >5-fold more resistant to docetaxel than were A549, H2228, and H3122 cells, which were exquisitely responsive (Appendix AA). Notably, EGFR-mutant PC-9/ER cells exhibited a prominent cross-resistance to docetaxel (Appendix AB).

When docetaxel was combined with nintedanib at a 1:1 ratio, an additive interaction was observed in most cell lines. When the relative amount of nintedanib in the combination was increased to achieve a 1:5 docetaxel:nintedanib ratio, most of the combinations were synergistic (Figure 2A). Such nintedanib-driven augmentation of the synergistic effects of the docetaxel:nintedanib combination was also evident in EGFR-TKI and ALK-TKI resistant derivatives (Appendix AC, *left*). When docetaxel was combined with silibinin, a strong synergism was observed in silibinin-resistant A549 cells at all tested ratios (1:10, 1:20, 1:100) of combined drugs. In nintedanib-resistant H1993 cells, the most efficacious combinations occurred at higher combination ratios (Figure 2B). In EGFR-TKI and ALK-TKI resistant derivatives, a higher number of synergistic interactions occurred at lower combination ratios of docetaxel and silibinin (Appendix AC, *right*).

### 2.5. Silibinin Suppresses Nintedanib-Unresponsive STAT3 Hyperactivation in NSCLC Cells

Ectopic overexpression and constitutive activation of STAT3 can prevent the tumoricidal effects of nintedanib on cancer cells [23,24]. Similarly, cells engineered to overexpress a constitutively active form of STAT3 remain largely unresponsive to the effects of silibinin [50]. We therefore speculated that, similar to other pathway-targeted cancer drugs [61], cell lines in which the nintedanib-driven blockade of the FGFR/SRC/STAT3 axis fails to fully suppress STAT3 activity could activate the JAK/STAT3 axis as a compensatory mechanism, which might be responsive to silibinin (Figure 3). To test this hypothesis, we first explored the correlation between the activation status of STAT3 and intrinsic responsiveness to nintedanib in our panel of NSCLC cell lines. Nintedanib-responsive PC-9 and A549 cells exhibited low constitutive levels of phospho-active STAT3^Tyr705^ that were fully suppressed in the presence of nintedanib (Appendix AA). Conversely, nintedanib-resistant H460 and H1975 cells exhibited a noteworthy constitutive hyperactivation of STAT3 that was fully unresponsive or further augmented in response to nintedanib (Appendix AA).

Once we established that STAT3 hyperactivation might be a key signaling alteration that contributes to primary/de novo nintedanib resistance in NSCLC cells, we explored whether the above-described capacity of silibinin to significantly rescue the sensitivity of NSCLC cells to nintedanib occurred upon silibinin-induced STAT3 inhibition. Immunoblot analyses in whole cell lysates confirmed that co-treatment with nintedanib and silibinin suppressed the residual STAT3 activity remaining after single-agent therapy in nintedanib-responsive PC-9 and A549 cells (Figure 3). Moreover, silibinin co-treatment for 24 h markedly suppressed nintedanib-unresponsive STAT3 hyperactivation in nintedanib-resistant H460 cells (Figure 3). The ability of silibinin to fully suppress nintedanib-unresponsive hyperactivation of STAT3 in nintedanib-resistant H1975 cells required a longer exposure (48 h; Figure 3).

To confirm that silibinin targeted the canonical role of STAT3 as a nuclear transcription factor, we examined the intracellular distribution of STAT3 and phospho-STAT3^Tyr705^ by immunoblot analysis of nuclear and cytosolic fractions of nintedanib-treated cells cultured in the absence or presence of silibinin (Appendix AB). Phospho-STAT3^Tyr705^ generated in nintedanib-sensitive PC-9 cells in response to IL-6 stimulation was found almost exclusively in the nucleus, a stimulatory effect that was fully prevented in the presence of silibinin (Appendix AB, *top*). In nintedanib-resistant H460 cells, constitutive immunoreactivity for phospho-active STAT3 was found almost exclusively in nuclear extracts and increased further in the presence of nintedanib. Silibinin treatment notably decreased constitutive activation of STAT3 and fully suppressed the nuclear hyperactivation of STAT3 occurring in the presence of nintedanib (Appendix AB, *bottom*). Consistent with the inhibition of STAT3 activation, silibinin markedly attenuated nuclear translocation of STAT3 in nintedanib-treated H460 cells. Immunofluorescent analysis confirmed the ability of silibinin to prevent cytoplasm-to-nuclear translocation and nuclear accumulation of phospho-active STAT3, regardless of the presence or absence of nintedanib (Appendix A).

### 2.6. Silibinin Prevents Lysosome Sequestration of Nintedanib in NSCLC Cells

Reversible sequestration and inactivation of nintedanib in acidic vesicles, a process known as lysosomal trapping [62,63,64,65,66], has been proposed as an intrinsic mechanism of resistance to nintedanib in FGFR-driven lung cancer cells [59,60]. This property prompted us to question whether silibinin might be capable of reversing lysosomal nintedanib sequestration.

We took advantage of the previous observation that nintedanib has autofluorescent properties [59], and we monitored its localization in NSCLC cells by light microscopy. Live-cell imaging revealed that upon nintedanib treatment, the drug accumulated in discrete structures in the cytoplasm of tumor cells. Co-staining of cells with LysoTracker^®^ Red—a fluorescent dye used for labeling acidic lysosomes in live cells—revealed a notable overlap with nintedanib staining, confirming the selective accumulation of nintedanib in lysosomes (Figure 4, *top panels*). We observed that nintedanib-resistant cell lines displayed a markedly greater accumulation of nintedanib than nintedanib-sensitive cells. Using flow cytometry, we established that the fluorescence signal produced by nintedanib was notably higher in nintedanib-resistant H460 and H1975 cells than in nintedanib-sensitive PC-9 and A549 cells (up to 30-fold; Figure 4, *bottom panels*).

Live cell fluorescence microscopy and flow cytometry analyses revealed that silibinin treatment promoted a significantly reduced lysosomal sequestration of nintedanib (up to 5- to 6-fold reduction) in nintedanib-resistant H460 and H1975 cells (Figure 5). No significant effects of silibinin were observed in nintedanib-sensitive PC-9 and A549 cells (Appendix A).

Exploiting the ability of the lipophilic weak base ammonium chloride (NH_4_Cl) to neutralize the acidic endosome-lysosome system as well as the well-known capacity of the specific inhibitor of vacuolar-type H^+^-ATPases bafilomycin A to directly inhibit lysosomal acidification, we explored to what extent the lysosomal alkalization sufficed to abrogate the lysosomal accumulation of nintedanib in nintedanib-resistant NSCLC cells. Silibinin treatment more closely resembled the ability of NH_4_Cl to partially abrogate the green fluorescence of nintedanib than that of bafilomycin A, which fully abolished the lysosomal accumulation of nintedanib (Appendix A). The proton capturing ability of the weak-base chloroquine to de-acidify the lysosome recapitulated the partial reversion of nintedanib lysosomal sequestration promoted by silibinin and NH_4_Cl (data not shown).

### 2.7. Clonogenic Assays and Real-Time Monitoring of Cell Proliferation Confirms the Synergistic Interaction between Silibinin and Nintedanib in NSCLC Cells

To assess the long-term effects of nintedanib and/or stress signals on cell survival, we tested the capacity of NSCLC cells to proliferate efficiently to form colonies. Clonogenic survival analyses revealed that nintedanib-resistant H460 cells failed to generate colonies long-term when cultured with the nintedanib plus silibinin combination. Co-treatment with silibinin fully suppressed colony formation in nintedanib-resistant H460 cells (Figure 6). Silibinin-resistant PC-9 cells failed to generate colonies long-term when cultured with the silibinin plus nintedanib combination (Figure 6).

A limitation of the MTT reduction-based assay is that because it is an end-point assay, it provides only a snapshot of a dynamic process. We therefore employed the impedance-based RTCA platform (xCELLigence), which is a label-free environment for cancer cells that can accurately inform about the characteristics of the response to treatment without the use of toxic/end-point assays, leading to the termination of the experiment. Using this platform, we captured real-time kinetic data on cell growth after treatment with the nintedanib/docetaxel combination in the absence or presence of silibinin (Appendix A, *top panels*). Cell proliferation rates and doubling times for PC-9, A549 and H460 cells cultured with or without nintedanib, docetaxel, silibinin and their respective combinations, were calculated as the slope of the growth curve of best fit from cell index recordings within a given time frame (i.e., between the 24 and 80 h interval). We found that co-treatment with silibinin augmented the ability of nintedanib to reduce the cell proliferation of PC-9, A549, and H460 cells. Accordingly, highly significant, supra-additive increases in cell doubling times were observed in those cells simultaneously exposed to nintedanib and silibinin (Appendix A, *bottom panels*).

### 2.8. The Nutraceutical Use of Silibinin Enhances the Clinical Response to Nintedanib/Docetaxel in Advanced Lung Adenocarcinoma

To assess the clinical impact of adding silibinin to the nintedanib/docetaxel combination in daily clinical practice, we carried out a retrospective, observational multicenter study assessing the efficacy of silibinin (5 capsules/day of Legasil^®^, equating to a 630 mg/day dose silibinin regimen [49,50]) in patients with advanced lung adenocarcinoma receiving the nintedanib/docetaxel combination in second- and further-line settings (Figure 7A).

Fifty-nine patients who started treatment between June 2014 and November 2017 were included in the study. Patients and tumor characteristics are summarized in Table 2. Forty-one (69.5%) patients received nintedanib plus docetaxel (control cohort) and 18 (30.5%) received the nintedanib/docetaxel combination and supplementation with Legasil^®^ (study cohort). In the whole cohort, the median follow-up was 20.2 months (interquartile range: 17.8–34.6). Most patients (78%) received nintedanib as second-line treatment and only 42% of patients received the recommended dose of docetaxel of 75 mg/m^2^. A significantly higher number of patients required a further dose reduction of docetaxel in the study cohort than in the control cohort (88.9% vs. 43.9%, *p* = 0.001).

The ORR was significantly higher in the study cohort than in the control cohort (55.6% vs. 22%; *p* = 0.011; Figure 7A). In the control cohort, no statistically significant differences were observed in the ORR according to *KRAS* status (wild-type: 22.2%, mutated: 25.0%, not evaluable: 18.2%, *p* = 1.000) or *EGFR* mutation (wild-type: 18.4%, mutated: 66.7%, *p* = 0.116). In the study cohort, no statistically significant differences were observed in the ORR according to *KRAS* status (wild-type: 50%, mutated: 25%, not evaluable: 70.0%, *p* = 0.320). All the patients included in the experimental cohort were EGFR wild-type, thus precluding the statistical analysis.

The median TTF was 2.8 months (95% confidence interval [CI]: 1.1–4.5) in the whole cohort, with no statistically significant differences between patients receiving second- or third-line treatment (*p* = 0.979). Median TTF was significantly higher in responders to nintedanib/docetaxel (6.0 months, 95%CI: 4.8–7.2) than in non-responders (1.8 months, 95%CI: 1.6–2.1) (*p* = 0.001; Appendix A). Compared with those receiving a reduced of dose docetaxel, patients receiving standard docetaxel dosing had higher median TTF (4.7 months 95%CI: 0.9–8.5 vs. 2.0 months 95%CI: 0.6–3.4, *p* = 0.006; Appendix A).

No significant differences were observed in the median TTF between the study and control cohorts in univariate analysis (4.7 months 95%CI: 2.8–6.6 vs. 2.4 months 95%CI: 1.8–2.9, *p* = 0.299; Figure 7B). In the subgroup of patients receiving a reduced dose of docetaxel (*n* = 34), the supplementation with Legasil^®^ was associated with a significant increase in median TTF (4.1 months 95%CI: 2.1–6.0 vs. 1.6 months 95%CI: 1.2–2.1, *p* = 0.007; Figure 7B). In the subgroup of patients receiving standard docetaxel dosing (*n* = 25), median TTF was 3.1 months (95%CI: 0.4–5.9) in the control cohort and 13.8–16.1 months in the two patients in the study cohort (no statistical analysis was performed because of the extremely small sample size). In multivariate analysis, both the supplementation with Legasil^®^ (hazard ratio [HR] 0.43, *p* = 0.013) and docetaxel dose reduction (HR 3.3, *p* < 0.001) were significantly associated with longer and shorter TTF, respectively (Table 3).

In the whole cohort, median OS since the beginning of first-line chemotherapy (OS1L) was 12.5 months (95%CI: 10.6–14.4) and median OS since the beginning of nintedanib/docetaxel combination (OS2L) was 6.8 months (95%CI: 4.6–8.9). There were no differences in median OS1L and OS2L between the study and control cohorts (13.2 months 95%CI: 3.5–22.8 vs. 12.4 months 95%CI: 10.4–14.4, *p* = 0.595; and 6.6 95%CI: 4–9.6 vs. 6.8 months 95%CI: 3.6–9.8, *p* = 0.877, respectively; Appendix A). No differences in median OS2L were observed between nintedanib/docetaxel-treated patients treated in second- or further-lines (*p* = 0.454). In the control cohort, OS2L was significantly higher in patients with *KRAS*-mutant than with *KRAS*-wild-type (10.1 months 95%CI: 0–22.9 vs. 7.3 months 95%CI: 2.7–11.9, respectively; *p* = 0.034; Appendix A), when patients with unknown *KRAS* mutational status were excluded from the analysis. Although a contrary trend was observed in the study cohort (3.0 months 95%CI: 0.3–6.0 in *KRAS*-mutant vs. 7.0 months 95%CI: 0.0–15.2 in *KRAS*-wild-type; *p* = 0.100), this trend in OS2L did not reach statistical significance (*p* = 0.100; Appendix A).

A significant increase of OS1L was observed in responders to first-line treatment compared with non-responders (14.1 months 95%CI: 4.3–23.8 vs. 11.4 months 95%CI: 9.6–13.3, *p* = 0.037). Patients achieving partial response to nintedanib/docetaxel showed also a higher median OS2L (10.9 months 95%CI: 6.8–15.1 vs. 5.4 months 95%CI: 4.7–6.1, *p* = 0.043). Patients with shorter TTF to first-line treatment (<9 months vs. > = 9 months) had shorter OS1L (11.2 months 95%CI: 9.7–12.7 vs. 29.6 months 95%CI: 21.1–38.2, *p* < 0.001) and OS2L (5.4 months 95%CI: 4.6–6.3 vs. 12.4 months 95%CI: 8.8–15.9, *p* = 0.013). In multivariate analysis, the sole variable that remained significant for OS2L was the reduced dosage of docetaxel in the first cycle (HR 2.282 *p* = 0.014) (Table 4).

## 3. Discussion

In the pivotal LUME-Lung 1 study [4], nintedanib in combination with docetaxel was found to improve the control of previously treated NSCLC disease and prolong overall and progression-free survival in patients with adenocarcinoma [4]. However, the lack of a clear improvement in the response rate, which is usually <10% in patients treated with docetaxel alone, has largely discouraged oncologists from using the nintedanib/docetaxel combination in an off-trial setting. Our present study provides a biological and clinical rationale for the addition of the flavonolignan silibinin to increase the efficacy of the nintedanib/docetaxel combination in patients with advanced NSCLC, for whom few effective second-line chemotherapy regimens are available.

Nintedanib is known to inhibit cell proliferation and induce apoptotic cell death in the three cell types contributing to angiogenesis: endothelial cells, pericytes, and smooth muscle cells [1]. Accordingly, the clinical benefit derived from nintedanib in patients with lung adenocarcinoma and fast-progressing tumors, and with primary or acquired resistance to chemotherapy, would suggest a bona fide anti-angiogenic functioning of the drug. However, resistance to nintedanib is common and patients ultimately relapse. In an attempt to elucidate the causes of recurrence, many studies have focused on tumor microenvironmental responses to the metabolic conditions induced by nintedanib, but we still lack strategies to target these mechanisms of resistance and clinically improve the efficacy of nintedanib. Importantly, we are now learning that the whole tumor system can develop resistance mechanisms in response to anti-angiogenic agents beyond those triggered by their anti-vascular effects [67,68,69]. To fully understand the resistance mechanisms to nintedanib, its anti-tumor effects must be elucidated both in the tumor vasculature and in the tumor cells themselves. In this regard, a potential direct impact of anti-angiogenic agents with multi-receptor TKI activity, such as nintedanib on tumor cell intracellular pathways, cannot be overlooked, as such downstream signaling pathways might be key to elicit the evasion adaptation of resistant cancer cells. By investigating how nintedanib acts on molecularly diverse NSCLC cells in terms of cytotoxicity, we found that those NSCLC cells with poorer responses to nintedanib exhibited increased activation levels of phospho-STAT3^Tyr705^ which were unresponsive to nintedanib. The suppression of nintedanib-refractory STAT3 hyperactivation by concurrent treatment with the STAT3i silibinin promoted synergistic anti-cancer effects. These findings might suggest that the refractoriness of NSCLC cells to nintedanib might rely, at least in part, on redundant or compensatory STAT3 signaling in tumor cells themselves. In FGFR-overexpressing cancer cells, tyrosine phosphorylation of STAT3 is also dependent on the concomitant FGFR-dependent activity of SRC and JAK2 kinase [70]. Since PDGFRβ has been reported to also induce the JAK2/STAT3 pathway by activating SRC, nintedanib might inhibit JAK2 by directly inhibiting PDGFRβ and SRC. Therefore, a multi-blockade of STAT3 activating events in response to nintedanib and silibinin appears to efficiently prevent NSCLC cells from escaping STAT3 inhibition. Nevertheless, our present findings suggest that direct inhibition of STAT3 activity with silibinin might represent a promising clinical strategy to circumvent NSCLC cancer cell-intrinsic nintedanib resistance.

Nintedanib is among the growing list of cancer drugs that can be sequestered in the lysosome [59,60], which reduces its therapeutic concentration in the cytosol. Nintedanib sequestration into the so-called lysosomal drug “safe-houses”—which results in an organelle-specific and pH-dependent nintedanib fluorescence—has been identified as an intrinsic mechanism of nintedanib resistance in FGFR-driven lung cancer cells [59]. Accordingly, treatment of NSCLC cells with chemicals capable of countering the lysosomal acidification such as chloroquine, which directly scavenges protons in the lysosomal lumen, and bafilomycin A1, which actively counteracts proton influx by H^+^-ATPase inhibition, which suppress the lysosomal sequestration of nintedanib and restore sensitivity to nintedanib [59]. We confirmed that massive lysosomal sequestration occurs in nintedanib-refractory NSCLC cells. Silibinin partially, but significantly, reduced the lysosomal entrapment of nintedanib in nintedanib-refractory NSCLC cells. The ability of silibinin to reverse lysosomal nintedanib sequestration in a similar way to the shift promoted by pharmacological agents which decrease acidifications by disrupting the ΔpH (i.e., NH_4_Cl and chloroquine) but not to those collapsing the whole ΔµH^+^ (bafilomycin A), likely reflects similarities and differences between their modes of action. Beyond lysosomotropic agents neutralizing the acidic endosome-lysosome system, possible strategies that might reverse lysosomal drug sequestration include alkalinizing agents, acid-labile conjugates, photodestruction and iron chelators, among others [59,60,71,72]. Although an alkalizing effect of silibinin has been reported in cancer cells [73], the molecular mechanism explaining the behavior of silibinin as a bona fide lysosome alkalizing small molecule is unclear. Silibinin is known to act as an iron chelator, even at acidic pH [74,75,76], and has been proposed as a chelation therapy for chronic iron overload [77,78]. Whether silibinin operates as a novel, metal-binding P-glycoprotein substrate like-drug that can be transported into lysosomes to trigger lysosomal membrane destabilization [60,79,80] and return nintedanib to the cytosol (Figure 5B), remains to be explored. Given that STAT3 enhances the lysosomal system [81] and directly associates with vacuolar H^+^-ATPase to regulate cytosolic and lysosomal pH [82], it is tempting to suggest that the STAT3 inhibitory activity of silibinin compromises lysosomal acidification, to exert synergetic growth inhibitory effects with nintedanib. However, it should be noted that the ability of lysosomal-associated STAT3 to maintain the alkaline cytosol and acidic lysosomal lumen occurs regardless of the activating SH2 binding site phosphorylation and DNA-binding activity of STAT3 [82]. Nevertheless, the significant prevention of subcellular lysosomal trapping, which is expected to increase cytosolic drug concentrations and, thus, the multikinase inhibition-based cytotoxic potential of nintedanib, represents an unforeseen mechanism through which silibinin could increase nintedanib availability at the target site and, consequently, circumvent lung cancer cell-intrinsic nintedanib resistance.

Most of the direct STAT3 targeted agents evaluated to date have been disappointing in the clinical arena due to suboptimal potency, unfavorable pharmacokinetic properties and other concerns over the relative lack of potency and selectivity [83]. While natural pharmacological inhibitors of STAT3 such as curcumin and butein have attracted attention because of their favorable toxicity profiles, their capacity to inhibit STAT3 phosphorylation, dimerization, acetylation and DNA-binding ability has been considered indirect and nonspecific [84]. Although the definition of an ideal STAT3 inhibitor for clinical use remains to be established, we took advantage of silibinin, which we have previously assessed in silico and experimentally validated with regards to its capacity to impair STAT3 activation [47,48,49,50]. Gain-of-function mutations computationally predicted to reduce the ability of silibinin to bind STAT3 with high affinity fully prevented its ability to inhibit STAT3 functionality, demonstrating the STAT3-dependency and largely eliminating the possibility that additional potential targets of silibinin might play a role in the biological actions [50]. Encouraged by these results, the lack of toxicity of silibinin and its oral bioavailability when provided as a commercially available nutraceutical (Legasil^®^), we evaluated its performance in a clinical study of 18 patients with lung cancer and brain metastases, which revealed its highly significant therapeutic activity, low toxicity, reversible secondary effects and compatibility with the standard-of-care [50]. The retrospective, observational multicenter study reported here assessed the efficacy of incorporating a nutraceutical supplementation of silibinin (5 capsules/day of Legasil^®^—630 mg/day dose) in patients with NSCLC receiving nintedanib/docetaxel combination in second- and further-line settings. The study cohort receiving silibinin benefited from an ORR (combined rates of complete and partial responses) greater than twice that observed in the control cohort. The patients supplemented with silibinin also benefited from a 2-fold decreased risk of treatment failure in multivariate analysis. Silibinin supplementation failed to provide significant benefit in terms of overall survival, which, in addition, was numerically lower than that originally reported in the LUME-Lung 1 study. A reason for this might be selection bias arising from the selection of a higher proportion of patients with poorer prognosis in the control and study groups, possibly resulting in non-random non-response. In this regard, our findings linking the sensitizing effects of silibinin to the lysosomotropic behavior of nintedanib might be clinically relevant. Thus, although the combination approach with silibinin might circumvent intrinsic nintedanib resistance through the lysosomal system, it might also distinctly alter the pharmacokinetic properties of nintedanib via (STAT3-dependent or -independent) modification of intracellular trafficking, autophagic activity, lysosomal load, lysosome biogenesis and/or lysosome-mediated cell death. Our laboratory is currently investigating how key characteristics of the lysosomal compartment in NSCLC cells, such as lysosome number, size and/or stability might impact on nintedanib responsiveness.

Since the publication of the LUME-Lung 1 trial [2], few data have been reported for the use of the nintedanib/docetaxel combination in NSCLC. The nintedanib Named Patient Use program suggested an encouraging efficacy of nintedanib/docetaxel combination following first-line platinum-based chemotherapy and subsequent immunotherapy in a real-world setting (*n* = 11) [3]. The prospective, noninterventional VARGADO study, which described data from a cohort of 22 patients who received nintedanib/docetaxel after progression on ICI therapy, highlighted the potential clinical benefit of rational treatment sequencing with nintedanib after progression on ICIs [4]. The prospective, multicenter, non-interventional LUMNE-BioNIS study, which has recently presented data from 67 patients with prior immunotherapy given in first- and later- lines, has shown that when used according to the approved label in routine practice, the nintedanib/docetaxel combination showed clinically relevant effectiveness [85]. Although we recognize that a major limitation of our current clinical study is its retrospective nature and the absence of randomization to each treatment intervention, it represents, to our best knowledge, one of the largest clinical series of patients with NSCLC treated with nintedanib/docetaxel without prior immunotherapy in everyday clinical practice reported thus far.

## 4. Materials and Methods

### 4.1. Cell Lines

The human NSCLC cell lines A549 (ATCC CCL-185), H460 (ATCC HTB-177), H1993 (ATCC CRL-5909) and H1975 (ATCC CRL-5908) were obtained from the ATCC (Manassas, VA, USA). H3122 (CVCL_5160) and H2228 (ATCC CRL-5935) cell lines, which harbor the E13:A20 and E6a/b:A20 variants of the EML4-ALK fusion, respectively, were made resistant to crizotinib (H3122/CR and H2228/CR) by incremental and continuous exposure to crizotinib, as described in [53,54]. PC-9 (RRID:CVCL_B260) cells, which harbor an EGFR activating mutation (Δ746–750), were made resistant to erlotinib (PC-9/ER) by incremental and continuous exposure to erlotinib, as described in [55,56]. PC-9, H460, and H1975 were made resistant to nintedanib (PC-9/NTD-R, H460/NTD-R, and H1975/NTD-R) by prolonged culture in graded concentrations of nintedanib. Parental PC-9 cells were obtained from the IBL cell bank (Gunma, Japan). All cells were routinely propagated in Dulbecco’s modified Eagle’s medium (DMEM) supplemented with 10% heat-inactivated fetal bovine serum (FBS; BioWhittaker Inc., Walkersville, MD, USA), 1% L-glutamine, 1% sodium pyruvate, 50 U/mL penicillin, and 50 μg/mL streptomycin. All cells were grown at 37 °C in a humidified atmosphere with 5% CO_2_ and were in the logarithmic growth phase at the initiation of the experiments. Cell lines were authenticated by STR profiling, both performed by the manufacturer and confirmed in-house at time of purchase following ATCC guidelines. Cells were passaged by starting a low-passage cell stock every month, up to 2–3 months after resuscitation. Cell lines were regularly screened for mycoplasma contamination using the MycoAlert Mycoplasma Detection Kit (Lonza, Verviers, Belgium).

### 4.2. Reagents

Nintedanib (BIBF 1120) and docetaxel were obtained from Selleck Chemicals LLC (Houston, TX, USA). Silibinin was purchased from Sigma-Aldrich (St. Louis, MO, USA). All reagents were dissolved in sterile dimethylsulfoxide (DMSO) to prepare 10 mmol/L stock solutions, which were stored in aliquots at −20 °C until use. Working concentrations were diluted in culture medium prior each experiment. Antibodies against total STAT3 (124H6, Cat. No 9139), phospho-STAT3 Tyr705 (D3A7, Cat. No 9145S), and phospho-STAT3 Ser727 (Cat. No 9134) were purchased from Cell Signaling Technology (Beverly, MA, USA). Antibodies against β-actin (Clone #2D4H5, Cat. No 66009-1-Ig,) and human recombinant human IL-6 (Cat. HZ-1019-10) were purchased from Proteintech Group, Inc., Rosemont, IL, USA). Lysotracker^®^ Red DND-99 (Cat. No L7528) was purchased from Thermo Fisher Scientific (Waltham, MA, USA). Ammonium chloride was purchased from Sigma-Aldrich (Cat. No A9434). Bafilomycin A1 was obtained from Calbiochem (Cat. No 196000).

### 4.3. MTT-Based Cell Viability Assays

For cell viability assays, NSCLC cells were seeded at 5 × 10^3^ cells/well in 100 µL of growth medium in 96-well plates. After overnight culture, cells were treated for 72 h with the indicated concentrations of each compound, combinations thereof or DMSO (*v*/*v*) (control wells). Compounds were not renewed during the entire period of cell exposure. For MTT assays, experimental media was replaced with fresh culture media (100 μL) and MTT (5 mg/mL in PBS) was added to each well at a 1/10 volume. After incubation for 3 h at 37 °C, the supernatants were carefully aspirated, 100 μL of DMSO was added to each well, plates were agitated to dissolve the crystal product and the optical density was measured at 570 nm using a multi-well plate reader. Cell viability in the presence of agents was reported as a percentage of the control cell optical density, which was obtained from control wells treated with the appropriate concentration of vehicle (DMSO) and processed simultaneously. For each treatment, cell viability was evaluated using the following equation: (OD_570_ of treated sample/OD_570_ of untreated sample) × 100.

Sensitivity of NSCLC cell lines to agents was expressed in terms of the concentration of drug required to decrease cell viability by 50% cell viability (IC_50_). Since the percentage of control absorbance was considered to be the surviving fraction of cells, the IC_50_ values were defined as the concentration of agents that produced 50% reduction in control absorbance and were estimated using non-linear regression analyses of dose-response curves.

### 4.4. Combination Index

The median effect analysis originally proposed by Chou and Talalay [57,58] was employed to determine the nature (synergism, additivity and antagonism) of drugs and drug interactions. Cells were treated with serial dilutions of each drug alone or with drug combinations at fixed ratios based on their corresponding IC_50_ values. The computed parameter, termed the combination index (CI), allows the quantitative determination of drug interactions at increasing levels of cell killing by classifying the tumoricidal activity as additive (CI value 1.0), synergistic (CI < 1.0), or antagonistic (CI > 1.0).

### 4.5. Immunoblotting

Cells were seeded in 6-well plates at 250,000 cells/well and allowed to grow overnight in maintenance cell culture media containing 10% FBS. The media were then replaced with DMEM containing 0.1% FBS with or without nintedanib and/or silibinin. Cells were incubated for a further 24 h, washed with ice-cold phosphate buffered saline (PBS), and scraped immediately after adding 30–75 µL of 2% SDS, 1% glycerol, and 5 mmol/L Tris-HCl, pH 6.8. The protein lysates were collected in 1.5 mL microcentrifuge tubes and samples were sonicated for 1 min (under ice water bath conditions) with 2 s sonication at 2 s intervals to fully lyse cells and reduce viscosity. Protein content was determined by the Bradford protein assay (Bio-Rad, Hercules, CA, USA). Sample buffer was added and extracts were boiled for 4 min at 100 °C. Equal amounts of protein were electrophoresed on 12% SDS-PAGE gels, transferred to nitrocellulose membranes and incubated with antibodies against STAT3 and phospho-STAT3 Tyr705, followed by incubation with a horseradish peroxidase-conjugated secondary antibody and chemiluminescence detection. β-actin (66009-1-Ig, Clone #2D4H5; Proteintech Group, Inc., Rosemont, IL, USA) was employed as control for protein loading.

### 4.6. Subcellular Fractionation

For immunoblotting of STAT3 and phospho-STAT3 Tyr705 in nuclear and cytosolic extracts, nuclei were purified using the Active Motif nuclear extract kit (Cat. No 40010 & 40410) according to the manufacturer’s protocol.

### 4.7. Immunofluorescence Microscopy

Cells seeded on gelatin-coated glass cover slips in a 24-well plate were fixed with 4% paraformaldehyde for 15 min, washed three times with ice-cold PBS, permeabilized by adding ice-cold 100% methanol and incubated with the respective antibodies against STAT3 (1:200 dilution) and phospho-STAT3 Tyr705 (1:1000 dilution). Antibody binding was localized with either a goat anti-rabbit IgG (H+L) secondary antibody, Alexa Fluor^®^ 594 conjugate or a goat anti-mouse IgG (H+L) secondary antibody, Alexa Fluor^®^ 488 conjugate (both from Invitrogen). Nuclei were counterstained with Hoechst 33342. Images were obtained with a Nikon Eclipse 50i fluorescence microscope including NIS-Elements imaging software.

### 4.8. Nintedanib Fluorescence

#### 4.8.1. Live Cell Fluorescence Microscopy

4 × 10^4^ cells were seeded in 12-well plates. After 24 h, cells were treated with graded concentrations of nintedanib and intracellular drug accumulation was imaged after 3 h on a live cell microscope (Nikon Eclipse Ts2) using a 20× objective equipped with a MicroScopia Digital XM Full HD Camera. To investigate the impact of lysosomal pH on the intralysosomal accumulation of nintedanib, cells were pretreated with 100 µmol/L silibinin, 10 mmol/L NH_4_Cl, 50 nmol/L bafilomycin A or 100 nmol/L LysoTracker^®^ Red prior to exposure to nintedanib. Images were merged using ImageJ software.

#### 4.8.2. Flow Cytometry

Nintedanib fluorescence was detected using 488 nm laser excitation wavelength (FITC channel) on a BD Accuri C6 Flow Cytometer. Data were analyzed using FCS Express 7 software (De Novo™ Software, Pasadena, CA, USA) and were depicted as mean fluorescence intensities (arbitrary units) of three independent experiments.

### 4.9. Real-Time Cell Growth Rate

Proliferation was measured using the xCELLigence Real Time Cell Analysis (RTCA) DP instrument (ACEA Biosciences, San Diego, CA, USA). NCSCL cells were plated at 5000 cells/well in 100 μL of fresh medium in an E-plate 16. Initial attachment and growth were continuously monitored for approximately 24 h at 37 °C and 5% CO_2_ for stabilization. Then, 100 μL of medium was removed from each well and replaced with fresh medium with or without drugs to achieve the appropriate final concentration. The plate remained in the RTCA Station for 96 h and impedance was monitored every 5 min for approximately 24 h at 37 °C and every 15 min for the next 72 h. Growth curves were plotted using the RTCA Software Package 1.2 (xCELLigence RTCA, Roche, Basel, Switzerland) and normalized to the time point of initial treatment; time-dependent cell index (CI), doubling time and slope graphs were generated as per the manufacturer’s instructions. Three biological replicates were evaluated in each experiment, which permits normalization to any time point, and results can be directly viewed in the software window. We conducted the normalization at one time point before the treatment.

### 4.10. Colony Formation Assays

Anchorage-dependent clonogenic growth assays were performed by initially seeding NSCLC cells into six-well plates at very low densities (~100 cells/well) and culturing in the presence or absence of graded concentrations of nintedanib and/or silibinin for 10 days (without refeeding) in a humidified atmosphere with 5% CO_2_, at 37 °C. Colonies were stained with crystal violet (0.5% *w*/*v*) in 80% methanol and 37% formaldehyde and the number of colonies with >50 cells/each were counted using ImageJ software.

### 4.11. Patients

We retrospectively reviewed all patients (*n* = 59) with NSCLC treated with the nintedanib/docetaxel combination in our institution (Catalan Institute of Oncology, Barcelona, Spain), which comprises three university cancer centers in Catalonia (Spain). All patients had adenocarcinoma histology and had received at least one previous systemic treatment. Patients treated between December 2014 and December 2015 were treated within the expanded compassionate program provided by Boheringer Ingelheim. Patients treated since January 2016 received treatments according to the approved indication by the Catalan Department of Health; that is, patients with advanced NSCLC adenocarcinoma histology whose tumors progressed within 9 months from the beginning of first-line treatment.

#### 4.11.1. Silibinin Regimen

Eighteen individuals received silibinin supplementation with Legasil^®^ (Mylan—Meda Pharma, Barcelona, Spain) at the Catalan Institute of Oncology, Hospital Universitari Dr. Josep Trueta of Girona (Spain). Each capsule of Legasil^®^ contains 210 mg Eurosil^85^ (60% of silibinin isoforms), which, according to the product patent data, has an increased release rate (80%) and improved absorbability. A titration was started with 2 capsules of Legasil^®^ (1-0-1) each day for the first three days of the plan and an additional capsule was then added until a 5 capsules dosage (2-2-1) was achieved or toxicity was observed. At the posology of five capsules per day of Legasil^®^, we provided 1050 mg of Eurosil^85^, which equated to a 630 mg-dose-a day silibinin regimen. Diarrhea was the sole drug-related adverse reaction that could lead to Legasil^®^ dose reduction. Once diarrhea resolved, treatment with Legasil^®^ was reinitiated at a lower dose.

#### 4.11.2. Outcomes Definitions

Tumor response was assessed using the Response Evaluation Criteria in Solid Tumors (RECIST), version 1.1. Efficacy analysis included all evaluable patients and the exploratory assessments analyzed were the following: overall response rate (ORR), defined as the combined rates of complete and partial responses; disease control rate (DCR), defined as the combined rates of complete response, partial response, and stable disease; time-to-treatment failure (TTF), defined as the time from date of first nintedanib/docetaxel treatment until first evidence of disease progression; overall survival (OS) since the beginning of first-line chemotherapy (OS1L), calculated from the first day of administration of first-line chemotherapy until the patient’s death or last date of follow-up; and overall survival since the beginning of nintedanib/docetaxel combination (OS2L), calculated from the first day of administration of the nintedanib/docetaxel combination until the patient’s death or last date of follow-up.

### 4.12. Statistical Analysis

#### 4.12.1. Cell-Based Assays

All observations were confirmed by at least three independent experiments performed in triplicate for each cell line and for each condition. Data are presented as mean ± SD. Two-group comparisons were performed using Student’s *t*-test for paired and unpaired values. Comparisons of means of ≥ 3 groups were performed by ANOVA, and the existence of individual differences, in case of significant F values at ANOVA, was tested by Scheffé’s multiple contrasts. *p* values < 0.01 and < 0.001 were considered to be statistically significant (denoted as * and **, respectively). All statistical tests were two-sided.

#### 4.12.2. Patients

Descriptive statistical analyses were conducted as appropriate. Kaplan-Meier survival curves were constructed and log-rank tests were used to compare survival between groups. Univariate and multivariate Cox regression analyses were conducted to compare the survival date. All tests were two-sided and *p* ≤ 0.05 was set as statistically significant. Statistical analyses were carried out using SPSS (release 2017, v25.0; IBM Corp., Armonk, NY, USA) and STATA (release 2013; StataCorp LP, College Station, TX, USA).

## 5. Conclusions

The dual activity of nintedanib as a TKI that targets not only angiogenesis in the tumor stroma, but also genetic alterations occurring in tumor cells such as those involving FGFRs and PDGFRs, has been suggested as an advantage over more selective anti-angiogenics. However, because numerous TKs share a common ability to induce downstream signaling effectors, nintedanib-treated NSCLC cells can engage feedback activation mechanisms such as STAT3, which are capable of promoting cell survival and limiting overall drug response in tumor cells themselves. Nintedanib is also among those TKIs that experience lysosomal drug sequestration as a tumor-cell intrinsic mechanism of multi-drug resistance that prevents them from reaching their targets. Here, we explored the molecular basis and the clinical benefit of incorporating silibinin—a flavonolignan extracted from milk thistle—into nintedanib-based schedules in advanced NSCLC. Molecular mechanisms dictating the cancer cell-intrinsic responsiveness to nintedanib such as STAT3 activation and lysosomal trapping were both amenable to pharmacological intervention with silibinin. The present in vitro data and clinical results should serve to accelerate the evaluation of the nutraceutical Legasil^®^ that contains a clinically relevant formulation of silibinin as an adjunct cancer treatment to the nintedanib/docetaxel combination. A prospective, powered clinical trial is warranted to confirm the clinical relevance of these findings in patients with advanced NSCLC.

## Figures and Tables

**Figure 1 cancers-13-04168-f001:**
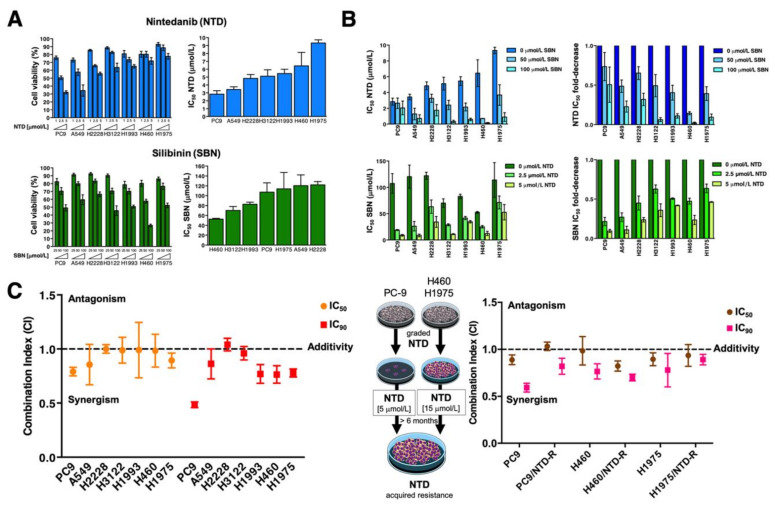
Analysis of the cytotoxic interaction between nintedanib and silibinin in NSCLC cells. (**A**) *Left panels.* The metabolic status of PC9, A549, H2228, H3122, H1993, H460, and H1975 NSCLC cell lines treated with increasing concentrations of nintedanib (NTD; 1.25, 2.5, and 5 µmol/L) and silibinin (SBN; 25, 50, and 100 µmol/L) was measured using MTT uptake assays, and cell viability was expressed as % uptake relative to untreated control cells (=100% cell viability). *Right panels.* Bar graphs of the IC_50_ values for each cell line calculated from the MTT assays as described in “Materials and methods”. The results are presented as the means (*columns*) ± S.D. (*bars*) (*n* = 3, in triplicate). (**B**) *Left panels.* Bar graphs showing the MTT-based IC_50_ values of NTD (*top*) and SBN (*bottom*) for each cell line calculated in the absence or presence of graded concentrations of SBN and NTD, respectively. *Right panels*. Bar graphs showing the fold-change in NTD (*top*) and SBN (*bottom*) IC_50_ values obtained in the absence or presence of graded concentrations of SBN and NTD, respectively. (**C**) Computed combination index (CI) values for the combination of NTD and SBN at 50% and 90% effect levels. CI values less than, equal to or greater than 1 indicates synergy, additivity or antagonism, respectively. The horizontal line at CI = 1 is the line of additivity. The results in A, B, and C panels are presented as the means (*columns*) ± S.D. (*bars*) (*n* = 3, in triplicate).

**Figure 2 cancers-13-04168-f002:**
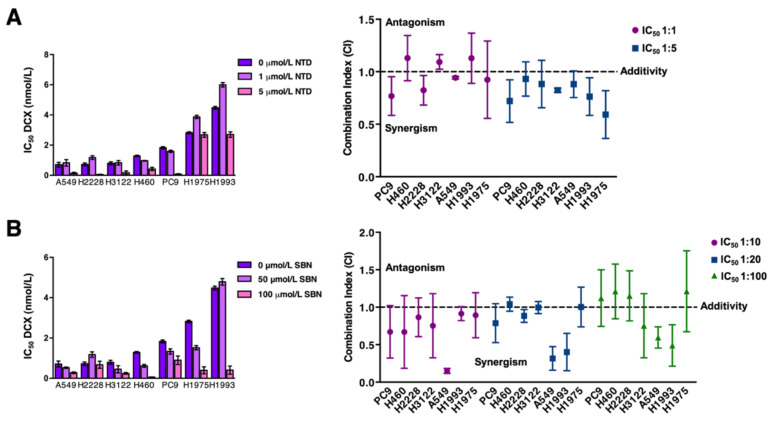
Analysis of the cytotoxic interaction between docetaxel, nintedanib, and silibinin in NSCLC cells. *Left panels.* Bar graph showing the MTT-based IC_50_ values of docetaxel (DCX) for each cell line calculated in the absence or presence of graded concentrations of NTD (**A**) and SBN (**B**). *Right panels*. Computed combination index (CI) values for the combination of DCX *plus* NTD (**A**) and DCX *plus* SBN (**B**) at 50% effect levels using different fixed ratio combinations of the drugs. CI values less than, equal to or greater than 1 indicates synergy, additivity or antagonism, respectively. The horizontal line at CI = 1 is the line of additivity. The results in A, B, C and D panels are presented as the means (*columns*) ± S.D. (*bars*) (*n* = 3, in triplicate).

**Figure 3 cancers-13-04168-f003:**
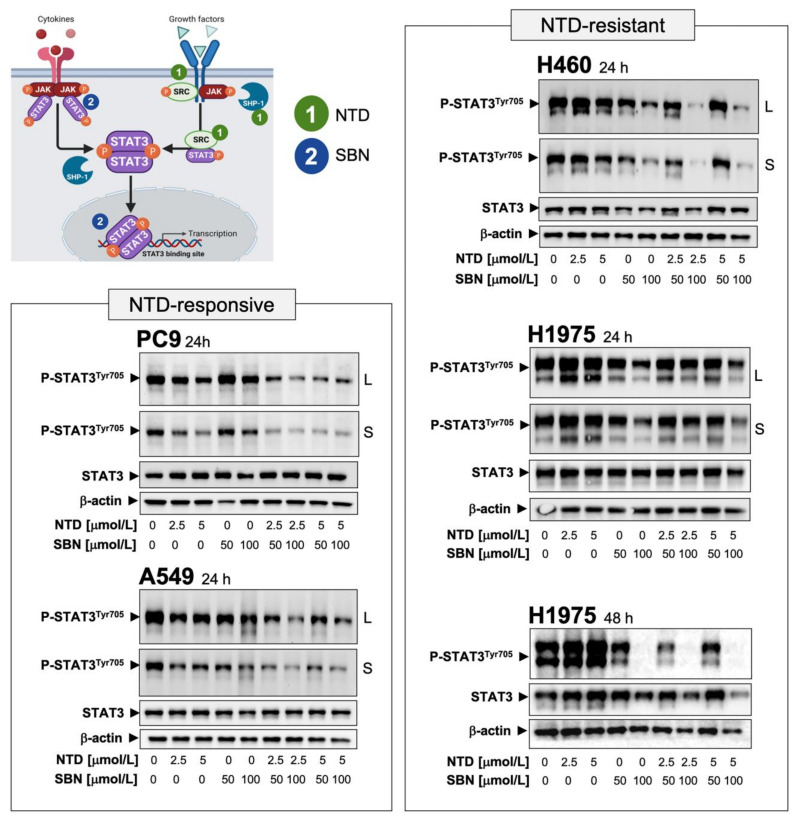
STAT3-targeted effects of the interaction between nintedanib and silibinin in NSCLC cells. Unstimulated STAT3 is activated by recruitment to phospho-tyrosine motifs existing within complexes of cytokine receptors (e.g., IL-6 receptor), growth factor receptors (e.g., EGFR, PDGFR, FGFR), or non-receptor tyrosine kinases (e.g., SRC) through its SH2 domain. STAT3 is then phosphorylated on the Tyr705 residue by activated tyrosine kinases (e.g., JAK, SRC) in receptor complexes. Phosphorylated STAT3 forms homodimers (and heterodimers with other STAT isoforms) and translocates to the nucleus, where STAT3 dimers bind to specific promoter elements of target genes to regulate gene expression via the STAT3 DNA-binding domain (DBD). Enhancing the activity of the SH2 domain-containing protein tyrosine phosphatase 1 (SHP-1) by the SHP-1 agonistic activity of nintedanib (NTD) can directly dephosphorylate STAT3 or its upstream JAKs to decrease the phospho-active STAT3 proteins (1). NTD might also inhibit STAT3 activity downstream of its direct actions on growth factor receptors such as PDGFRβ and FGFR or non-receptor tyrosine kinases such as SRC (1). Silibinin (SBN) can directly bind to the (JAK and SRC-activated) SH2 domain of STAT3 to prevent Y705 phosphorylation-related STAT3 activation and dimerization (2). SBN can also establish direct interactions with DNA in its targeting to the STAT3 DBD (2), thereby preventing the binding of STAT3 to its consensus DNA sequence. Dual blockade of the STAT3 activating events by combining NTD and SBN might efficiently prevent NSCLC cells from escaping from STAT3 inhibition in response to NTD/SBN as single agents, thereby providing a basis for a molecular rationale for the incorporation of SBN into nintedanib-based schedules in NSCLC. NTD-responsive PC-9 and A549 cells, and NTD-resistant H460 and H1975 cells were serum-starved overnight and then left untreated or treated with NTD in the absence or presence of SBN for 24 or 48 h. Levels of phospho-STAT3^Tyr705^ and STAT3 were detected by immunoblotting using specific antibodies (S: short exposure; L: long exposure). Figure shows representative immunoblots of multiple (*n ≥* 5) independent experiments. The uncropped blots and molecular weight markers are shown in Appendix A.

**Figure 4 cancers-13-04168-f004:**
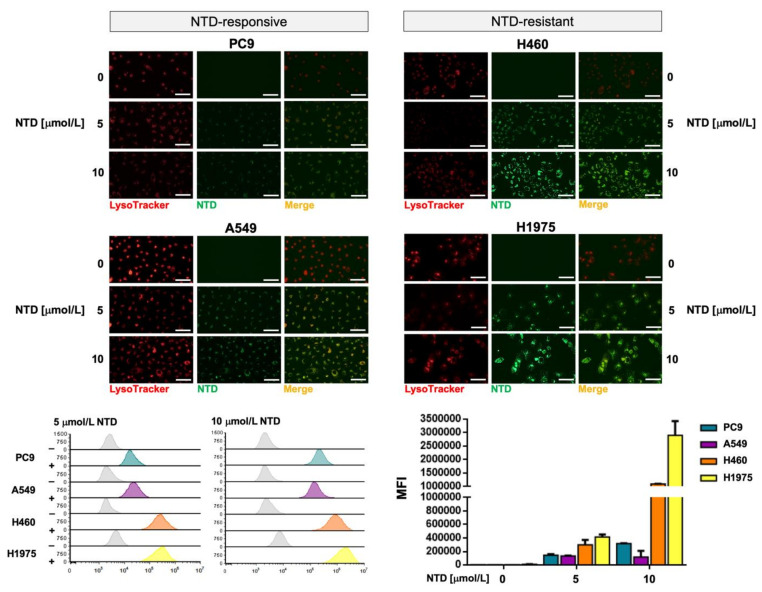
Lysosomal sequestration of nintedanib in NSCLC cell lines. *Top.* Subcellular distribution of 5 and 10 µmol/L nintedanib in PC-9, A549, H460 and H1975 cells after 3 h drug exposure was analyzed by live cell fluorescence microscopy in the FITC channel. LysoTracker^®^ Red served as a marker for the lysosomal compartment. Nintedanib/LysoTracker^®^ spatial overlap is also shown. The scale bar indicates 50 µm. *Bottom.* Dose-dependent green fluorescence activity of nintedanib (NTD) in NTD-responsive PC-9 and A549 cells and in NTD-resistant H460 and H1975 cells was analyzed by flow cytometry. Each experimental value represents the mean NTD-associated fluorescence (*columns*) ± S.D. (*bars*) of 3 independent experiments. (MFI: Median Fluorescence Intensity).

**Figure 5 cancers-13-04168-f005:**
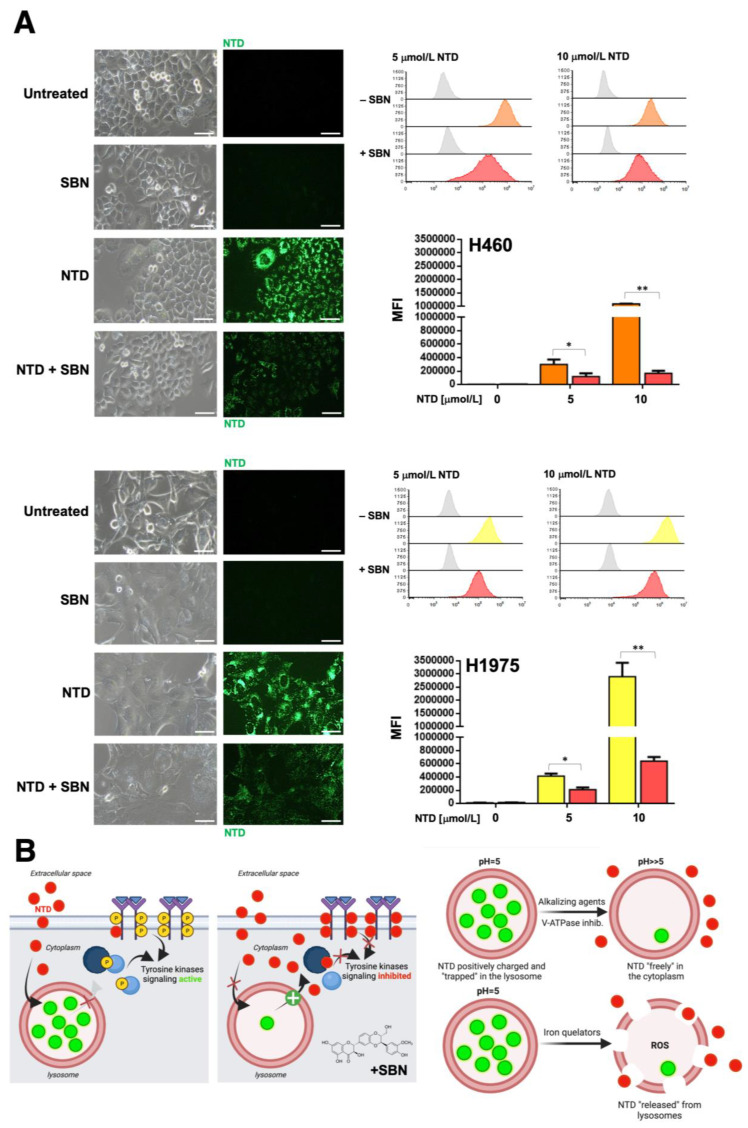
Effects of silibinin on the lysosomal sequestration of nintedanib in nintedanib-resistant NSCLC tumor cells. (**A**) The impact of 100 µmol/L silibinin (SBN; 1 h pre-treatment) on the intracellular accumulation of 5 and 10 µmol/L nintedanib (NTD) was analyzed by live cell fluorescence microscopy (*left panels*) and flow cytometry (*right panels*) after 3 h drug exposure. Each experimental value represents the mean NTD-associated fluorescence (*columns*) ± S.D. (*bars*) of 3 independent experiments. The scale bar indicates 50 µm. Comparisons of means were performed by ANOVA. *p* values < 0.01 and < 0.001 were considered to be statistically significant (denoted as * and **, respectively; n.s. not significant). (**B**) Sequestration of NTD into lysosomes provides a mechanism of NTD resistance in NSCLC cells. Overcoming NTD trapping by alkalizing lysosomes (e.g., by using NH_4_Cl, chloroquine, bafilomycin A) or disrupting lysosomes containing sequestered NTD (e.g., by iron chelators) can potentiate the effects of NTD treatment. (MFI: Median Fluorescence Intensity).

**Figure 6 cancers-13-04168-f006:**
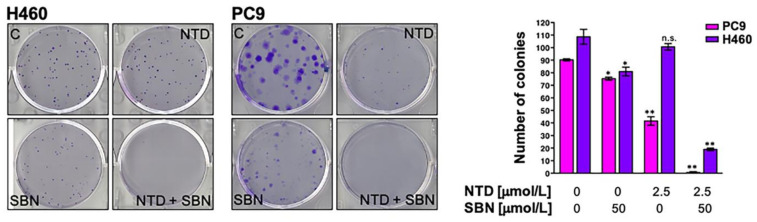
Analysis of clonogenic survival in response to nintedanib plus silibinin. *Left.* Representative images from 6-well plates of colonies of H460 and PC-9 cells treated with nintedanib and/or silibinin. ImageJ (NIH) was used to quantify the number of 7-day-old colonies stained with crystal violet. *Right. Columns* and *error bars* represent mean values and S.D., respectively. Comparisons of means were performed by ANOVA. *p* values < 0.01 and < 0.001 were considered to be statistically significant (denoted as * and **, respectively; n.s. not significant).

**Figure 7 cancers-13-04168-f007:**
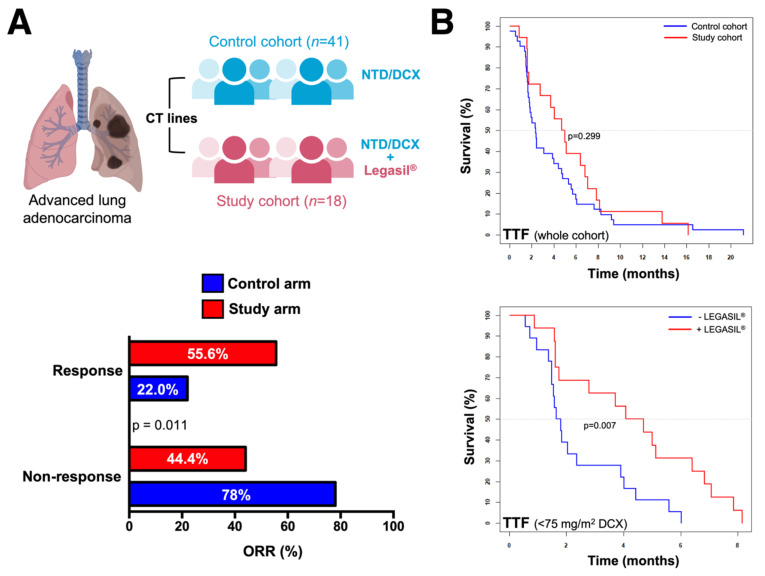
Impact of the silibinin-containing nutraceutical Legasil^®^ on the clinical efficacy of the nintedanib/docetaxel combination in advanced lung adenocarcinoma. (**A**) *Top.* We conducted a retrospective, observational multicenter study to assess the efficacy of incorporating the oral nutraceutical product Legasil^®^ containing silibinin in patients with advanced lung adenocarcinoma receiving nintedanib/docetaxel combination in second- and further-line settings (*n* = 59). *Bottom.* Overall response rate (ORR, in %)—defined as the combined rates of complete and partial responses–-in the control and study arms. (**B**) Kaplan–Meier TFF curves of patients stratified by control/study arms (*t**op*) and (−) Legasil^®^/(+) Legasil^®^ co-treatment (*bottom*).

**Table 1 cancers-13-04168-t001:** NSCLC cell lines characteristics.

Cell Line	Mutational Features
**A549**	EGFR WT; KRAS mutant (G12S)
**H460**	EGFR WT; KRAS mutant (Q61H)
**H1993**	EGFR WT; MET amplification
**H3122**	EGFR WT; EML4-ALK variant 1
**H2228**	EGFR WT; EML4-ALK variant 3
**H1975**	EGFR mutant (L858R/T790M); KRAS WT
**PC9**	EGFR mutant (DelE746-A750)

**Table 2 cancers-13-04168-t002:** Baseline demographics of the cohorts and tumor characteristics.

	Study Cohort	Control Cohort	
	(*n =* 18)	(*n* = 41)	*p*-Value
**Sex**			0.187
Male	12 (67%)	34 (83%)	
Female	6 (33%)	7 (17%)	
**Age**			0.490
Mean ± SD (range)	61.2 ± 10.3 (43–79)	59.5 ± 7.3 (45–71)	
**Line of treatment**			1.000
2nd line	14 (78%)	32 (78%)	
≥3rd line	4 (22%)	9 (22%)	
**TTF 1st line (months)**			0.439
Median (p25,p75)	4.7 (4.1, 6.5)	4.4 (1.9, 7.6)	
**Response to 1st line**			0.188
Response (PR)	5 (27.8%)	13 (31.7%)	
Stable disease	9 (50.0%)	11 (26.8%)	
Progression disease	4 (22.2%)	17 (41.5%)	
**Initial docetaxel dose**			0.001
75 mg/m^2^	2 (11%)	23 (56%)	
<75 mg/m^2^	16 (88.9%)	18 (44%)	
**ECOG PS**			0.144
0	5 (27.8%)	3 (7.3%)	
1	13 (72.2%)	36 (87.8%)	
2	0 (0%)	2 (4.9%)	
**EGFR status**			0.546
*EGFR*-mutant	0 (0%)	3 (7.3%)	
*EGFR*-wild-type	18 (100%)	38 (92.7%)	
**KRAS status**			0.099
*KRAS*-mutant	4 (22.2%)	12 (29.3%)	
*KRAS-*wild-type	4 (22.2%)	18 (43.9%)	
*KRAS*-not evaluable	10 (55.6%)	11 (26.8%)	
**PD-L1 status**			0.222
0%	3 (16.7%)	11 (26.8%)	
1–49%	1 (5.5%)	5 (12.2%)	
≥50%	3 (16.7%)	1 (2.4%)	
Not evaluable	11 (61.1%)	24 (58.6%)	

TTF: Time-to-treatment failure; PR: Partial response; ECOG PS: Eastern Cooperative Oncology Group Performance status; PD-L1: Programmed death-ligand 1.

**Table 3 cancers-13-04168-t003:** Univariate and multivariate Cox proportional analysis for TTF.

	Univariate Analysis			Multivariate Analysis		
	HR	(95%CI)	*p*-Value	HR	(95%CI)	*p*-Value
**Age (years)**						
≤65	1	(referent)		1	(referent)	
>65	0.60	(0.33–1.10)	0.096	0.74	(0.39–1.37)	0.335
**Sex**						
Female	1	(referent)		1	(referent)	
Male	1.06	(0.57–1.98)	0.847	0.72	(0.47–1.70)	0.890
**Docetaxel**						
75 mg/m^2^	1	(referent)		1	(referent)	
<75 mg/m^2^	2.23	(1.18–4.22)	0.013 *	3.30	(1.69–6.45)	<0.001 *
**Legasil^®^**						
No	1	(referent)		1	(referent)	
Yes	0.74	(0.42–1.31)	0.302	0.43	(0.22–0.84)	0.013 *

HR: hazard ratio; CI: confidence interval; * statistically significant (*p* < 0.05).

**Table 4 cancers-13-04168-t004:** Univariate and multivariate Cox proportional analysis for OS2L.

	Univariate Analysis			Multivariate Analysis		
	HR	(95% CI)	*p*-Value	HR	(95% CI)	*p*-Value
**Age (years)**						
≤65	1	(referent)		1	(referent)	
>65	1.05	(0.56–1.95)	0.886	1.07	(0.58–2.00)	0.823
**Sex**						
Female	1	(referent)		1	(referent)	
Male	1.26	(0.65–2.45)	0.503	1.19	(0.60–2.32)	0.621
**Docetaxel**						
75 mg/m^2^	1	(referent)		1	(referent)	
<75 mg/m^2^	1.80	(1.02–3.15)	0.041 *	2.22	(1.18–4.17)	0.014 *
**Legasil^®^**						
No	1	(referent)		1	(referent)	
Yes	0.95	(0.52–1.75)	0.878	0.63	(0.32–1.25)	0.190

HR: hazard ratio; CI: confidence interval; * statistically significant (*p* < 0.05).

## Data Availability

The data that support the findings of this study are available from the corresponding authors, upon reasonable request.

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
