# Peer review of "Silibinin Suppresses Tumor Cell-Intrinsic Resistance to Nintedanib and Enhances Its Clinical Activity in Lung Cancer"

_cancers, 2021, doi:10.3390/cancers13164168_

Round 1

Reviewer 1 Report

The revised version has significantly improved the original one. In particular, all my comments have been addressed and I have no further concerns.

This manuscript is a resubmission of an earlier submission. The following is a list of the peer review reports and author responses from that submission.

Round 1

Reviewer 1 Report

This article studies the possible activity of the STAT3 inhibitor silibinin in advanced non-small cell ling cancer (NSCLC). The first part of the study is made on a number of NSCLC cell lines to determine the tumoricidal interactions between nintedanib and silibinin. Data on the tumoricidal interations between nintedanib, docetaxel and silibinin are also presented. These studies are very detailed and provide solid evidence that cells with worse responses to nintedanib presented persistent STAT3 activation that was reversed upon silibinin treatment. Therefore, for some cell lines nintedinib and silibinin showed a synergistic effect on cell toxicity. The second part of the article consists of a retrospective, observational multicentre study to determine the efficacy of silibinin administration to advanced NSCLC patients being treated with nintedanib/docetaxel. Silibinin is administered in these patients as a nutraceutical product. The results obtained show improved overall response rate and higher time to treatment failure in patients receiving silibinin. This study is promising and very interesting but the number of patients studied is still small and must be confirmed in a future prospective powered clinical trial, as stated by the authors.

I consider that the article is of great interest in the NSCLC field. It is very well presented and the data support the conclusions of the authors. There are some points where the article could be improved, as follows:

Major points:

  1. The interactions between docetaxel and silibinin, shown in Figures 2B and S2C (right panel) are not described in the text.
  2. The experiments shown in Figure 3B for the H460 cell line seem to indicate that STAT3 expression is strongly repressed by sibilinin specifically in this cell line. Is this observation correct and reproducible? In that case, the authors should discuss this observation and the possible mechanisms involved.
  3. The results shown in Figure 3B for A549 show that STAT3 phosphorylation is strongly inhibited by 100 mm/L of silibinin. However, this inhibition is smaller when cells are co-treated with nintedanib, either 2.5 or 5 mm/L . Could the authors discuss of a possible explanation?

Minor points:

- Line 81, “have largely” is repeated twice.

- Lines 293-411 and 638-641. The words “OS1L”/“OSL1” and “OS2L”/“OSL2” are used at different points and should be unified.

- Lines 612-613, the authors indicate that a group of patients received treatments according to the approved indication by the Catalan Department of Health. This treatment should be specified.

Reviewer 2 Report

cancers-954626: 

Major comments: 
1. In vitro studies: 
- Lack of novelty findings as the data is quite simple, 
Additionally, there is problematic about used concentrations. 
Just for example, as shown in Fig 3B, the beta-actin was lost expression itself, indicating the nature toxic for cell treatment and changed of cell morphology
- As summarized in Fig 3A, STAT3 phosphorylation works mainly inside nucleus for transcriptional function while the group only tested the whole of cell lysate.
2. Clinical testing, 
This is a good point of this study. However, the studied cohorts were too small while there are several comparations without statistically significant differences as noted by the group.